# A Simple Method to Estimate Entropy and Free Energy of Atmospheric Gases from Their Action

**DOI:** 10.3390/e21050454

**Published:** 2019-05-01

**Authors:** Ivan Kennedy, Harold Geering, Michael Rose, Angus Crossan

**Affiliations:** 1Sydney Institute of Agriculture, University of Sydney, Sydney, NSW 2006, Australia; 2QuickTest Technologies, PO Box 6285 North Ryde, North Ryde, NSW 2113, Australia; 3NSW Department of Primary Industries, Wollongbar, NSW 2447, Australia

**Keywords:** statistical mechanics, partition functions, translational entropy, rotational entropy, vibrational entropy, Gibbs and Helmholtz energies

## Abstract

A convenient practical model for accurately estimating the total entropy (Σ*S*_i_) of atmospheric gases based on physical action is proposed. This realistic approach is fully consistent with statistical mechanics, but reinterprets its partition functions as measures of translational, rotational, and vibrational action or quantum states, to estimate the entropy. With all kinds of molecular action expressed as logarithmic functions, the total heat required for warming a chemical system from 0 K (Σ*S*_i_*T*) to a given temperature and pressure can be computed, yielding results identical with published experimental third law values of entropy. All thermodynamic properties of gases including entropy, enthalpy, Gibbs energy, and Helmholtz energy are directly estimated using simple algorithms based on simple molecular and physical properties, without resource to tables of standard values; both free energies are measures of quantum field states and of minimal statistical degeneracy, decreasing with temperature and declining density. We propose that this more realistic approach has heuristic value for thermodynamic computation of atmospheric profiles, based on steady state heat flows equilibrating with gravity. Potentially, this application of an action principle can provide better understanding of emergent properties of many natural or evolving complex systems, including modelling of predictions for global warming.

## 1. Introduction

As defined by Clausius [1], entropy can be considered as measuring the “self-reservoir of heat required to raise the temperature of a system of ideal gas molecules to the temperature *T*”. Thus, in agreement with the third law of thermodynamics, at the absolute zero of temperature Kelvin (K), the entropy should also be zero.

Clausius named entropy using the Greek word for ‘in-turning’ or transformation, a dynamic definition that this paper will show is highly apt. As a property of state (pressure, temperature, and volume) and irrespective of the path undertaken by a system of molecules to arrive at that condition, entropy is therefore an important feature of atmospheric gases. Indeed, its capacity to explain how thermal radiation may be absorbed and partitioned into different degrees of freedom of motion is the key information that explains the warming potential of greenhouse gases. 

The total entropy of an ideal gas molecule can be calculated as the sum of terms,
*S*_Total_ = *S*_t_ + *S*_r_ + *S*_v_ + *S*_e_ + S_n_ + ….,
where the subscripts refer to translational, rotational, vibrational, electronic, and nuclear entropy terms, respectively. To estimate the total thermal energy needed to reversibly heat (i.e., without doing other work) a system of gas molecules, we need only multiply by the temperature *T*. This thermal energy can be thought of as both the kinetic and potential energy contained in the field of the molecular system [2,3]. Furthermore, this thermal energy input is physically required to sustain the physical action of the system [4].

This paper seeks to place thermodynamics within easier reach of non-specialists by giving a key role to action, a physical property that is, realistically, the focus of our interest. Action is related to the vector and angular momentum, with similar dimensions of mass by velocity by inertial radius (*mvr*), but is a distinct scalar quantity independent of direction. Like entropy, it is an extensive or cumulative property, but with physical dimensions of the integral of energy with time, or of the instantaneous angular momentum with respect to angular motion; classically, action was considered as the integral of momentum with distance. As a variable property of conservative systems, action has been considered to take stationary values, a result sometimes referred to as the principle of least action. In fact, all these viewpoints of action are equivalent. Note that angular motion is the ratio of circumference to radius and is thus physically dimensionless, although we still measure it in degrees or radians.

To illustrate the utility of action theory, we advance a unified model that we will show is valid for calculating the entropy of atmospheric gases. In the range of ambient temperatures relevant to the Earth’s atmosphere, this realistic model gives results for absolute entropy that are consistent with previous experimental data. This action model may help provide an approach to prediction of the rate of global warming based on causal responses to the increasing greenhouse gas content of the atmosphere, rather than statistical correlations. In particular, this aid to exploring a direct relationship between thermodynamics and gravity may provide a dynamic view of how the thermal properties of the atmosphere have significance for warming and climate change.

## 2. Materials and Methods

Here, we show how the classical formulae for estimating translational, rotational, and vibrational entropy using partition functions may be reviewed as physical action. Although entropy data for atmospheric gases are readily available in standard tables [5,6], the methods developed here illustrate how easily such data can be manipulated to account for real environmental conditions.

Since they are relevant to the following methods, we state, here, the partition functions for ideal gases that have been used to calculate the entropies for molecular translation, rotation, and vibration [5]. These functions are given in all standard texts (e.g., Moore [6]) on modern statistical mechanics, the discipline founded jointly by Ludwig Boltzmann [7] and J. Willard Gibbs [8]. The factors governing these functions include absolute temperature (*T* in K), Boltzmann’s constant (*k =* 1.3806 × 10^−23^ J·K^−1^), Planck’s quantum of action (*h =* 6.626 × 10^−34^ J·sec) and the system volume (*V*) and moments of rotational inertia (*I*). For ease of use and consistency in dimensions, all modelling and calculations have been performed in centimetre–gram–second (cgs) units, before conversion to Systéme Internationale (SI) units where required.


*Translational partition function*
*Q*_t_ = (2π*mkT*/*h*^2^)^3/2^*V*(1)


Here, *V* is taken as the system volume occupied by the molecules [5].


*Rotational partition function (linear molecule)*
*Q*_r_ = 8π*IkT*/*h*^2^(2)



*Rotational partition function (non-linear molecule)*
*Q*_r_ = 8π^2^(8π^3^*I_A_I_B_I_C_*)^1/2^(*kT*/*h*^2^)^3/2^(3)


*Vibrational partition function (polyatomic molecules)**Q*_vi_ = ∏_i_ [1 − exp^−*hν*/*kT*^]^−1^,(4)
where ∏_i_ indicates a product of i functions, for each mode of vibration.

It will be shown that these functions can all be considered as dimensionless statistical measures of relative molecular action, using Planck’s quantum *h* or its reduced form per radian, *ħ,* as a natural reference unit. Note that there are only three sources of variation in the partition functions—inertial mass; temperature or the root mean square velocity; and pressure or density. Even the exponential partitions for vibrational function can be rendered as action ratios when elevated energy states are recognised as inversely proportional to number density, a surrogate for probability. Then, a vibrational state of low probability also has an equivalent low number density, giving it a high action value with large inertial radius.

### 2.1. Translational Entropy and the Sackur–Tetrode Equation

The Sackur–Tetrode equation was published early in the 20th century [5], based on Gibbs’ theory of statistical mechanics [8].
*S_t_ = R*[ln(2*πmkT*)^3/2^V/*h*^3^N + 2.5](5)

This equation allows calculation of the translational entropy of N molecules of an ideal monatomic gas. This is derived from the relationship from the calculus of total entropy, including translational *S*_t_ and internal *S*_int_ parts as follows [5].
*S* = *RT*(∂lnQ/∂T)_V_ + *R*lnQ − klnN! = *S*_t_ + *S*_int_(6)
*S* = *RT*[(∂lnQ_tr_/∂*T*) + (*d*lnQ_int_/*dT*)] + R[lnQ_tr_ + lnQ_in_] – *k*lnN!(7)

Here the factor *k*lnN! allows for the inability to distinguish between N identical molecules.

For ideal monatomic gases, no internal entropy (from rotation or vibration) at normal temperatures exists, so the differential of the internal partition function Q_int_ in the above equation can be ignored.
*S_t_* = *RT*[(∂lnQ_tr_/∂*T*)] + R[lnQ_tr_] − *k*lnN!(8)

Using the solution from the Schrodinger equation, that *Q*_t_ is (2π*mkT*/*h*^2^)^3/2^*V* as given in Equation (1), and that the Stirling approximation for lnN! is effectively NlnN−N, we have from [5]:*S_t_* = 3/2*R* + *R*ln[(2π*mkT*/*h*^2^)^3/2^*V*] − *R*ln(N − 1) = *R*[ln(2*πmkT*)^3/2^V/(*h*^3^N) + 2.5].(9)

Note the close similarity of Equation (9) to the translational partition function. This is a well-known result of statistical mechanics, with *h* being the constant introduced by Planck as the quantum of action for radiation. Despite their lack of rest mass, all energy quanta possess an action of magnitude *h* and their energy is given by *hν*, where *ν* is their frequency.

Here, we introduce a revised approach, based on the use of the property of state that action (@) is equal in magnitude to the product (*mv* × *r*) of molecular momentum (*mv*) by a radial parameter (*r*) [2,3]. This allows the establishment of the relative action, a ratio or pure number suitable for logarithmic expression. Some expressions of entropy in text books include isolated terms such as the logarithm of the temperature ln*T*. Strictly, this is invalid, as logarithms can only be taken of pure numbers or dimensionless ratios, and never of quantities with physical dimensions. 

For each of the three translational degrees of freedom, the translational action can be derived from the kinetic energy, given each has ½*kT* of kinetic energy. For three degrees of freedom, we can estimate the action, *mvr* or *I_t_ω_t_*, as follows. The three-dimensional kinetic energy (½*mv*^2^) for motion with polar coordinates are shown in Equation (10), with translational angular velocity (*ω*_t_ or *dӨ/dt*) given in radians per second.
½*mv*^2^ = 3/2*kT* = ½*mr*_t_^2^*ω*_t_^2^ = ½*I*_t_*ω*_t_^2^(10)
*I*_t_*ω*_t_ = 3*kTI*_t_*/ω*_t_ = (3*kTI*_t_)^1/2^(11)

The mean translational action *@*’_t_ is thus defined as equal to (*3kTI_t_*)*^1/2^,* although a correction factor—as indicated by the prime and discussed below—is required because 3*kT* is a statistical result from the three-dimensional Maxwell distribution, equal to twice the most probable kinetic energy ½*mv*^2^ for the root mean square velocity *v* in three dimensions. Moreover, 50% of molecules have speeds greater than the root mean square velocity, which is 1.085 times the mean speed of the ideal gas molecules ([9], Table 8). In the Maxwell distribution, the most probable velocity is slightly less than either of these speeds.

We can regard the system volume *V* as containing N cubic cells of volume *a*^3^, a cell for each gas molecule. Then, for *r*_t_ arbitrarily taken as the mean value of the half-distance between the centres of any two nearest neighbour gas molecules, *a*^3^ is equal to (2*r*_t_)^3^ or 8*r*_t_^3^. Considering a mole of gas at 298.15 K and 1 atmosphere pressure (N = 6.022169 × 10^23^ molecules in 24465.1 mL), then *r*_t_ or (*V/*N)^1/3^/2 is equal to 1.7188 × 10^−7^ cm. 

We can then substitute into the Sackur–Tetrode Equation (5). Taking the *r_t_^3^* term inside the brackets, we have
*S*_t_ = *R*ln[8e^5/2^(2π*mr*_t_^2^*kT*)^3/2^/*h*^3^].(12)

Then, *mr*_t_^2^*kT* is taken as equal to *kTI*_t_ and @’_t_, an uncorrected version of the translational action equal to (*3kTI*)^1/2^.
*S*_t_ = *R*ln[8(2π/3)^3/2^e^5/2^(@’_t_/*h*)^3^](13)

For *ħ* equal to *h*/2π, the reduced quantum of action, we have,
*S_t_* = *R*ln[*e**^5/2^*(*2/3π*)*^3/2^*(*@’*_t_/*ħ*)^*3*^](14)

If the factor of (2/3π)^3/2^ were to be incorporated into the action term, the inertial radius would be decreased to 7.92287 × 10^−8^ cm rather than 1.7188 × 10^−7^ cm. However, for computation, we initially assume a translational resonance symmetry factor 1/z_t_ of 1/10.22967, replacing (2/3π)^3/2^. In a preprint lodged in Cornell’s arXiv [10], a z_t_ factor was retained; more recently, we have identified this z_t_ constant as being exactly equal to the inverse of (2 × 1.0854)^3^, providing two corrections to prevent double counting in neighbouring molecular couples, and for the ratio of the root mean square velocity to the mean velocity respectively [11], as noted above. This allows for calculation of a mean translational action value, *n*_t_, corresponding to a quantum field state for the molecule, equilibrated with gravity.
*S_t_* = *R*ln[*e**^5/2^*(*@’*_t_/*ħ*)^*3*^/z_t_] = *R*ln[*e**^5/2^*(*@*_t_/*ħ*)^*3*^] = *R*ln[*e*^*5/2*^(*n*_t_)^*3*^](15)

Given the disorderly nature of the spatial distribution of the translating particles (see Figure 1), the translational quantum state factor *n* cannot have a precise continuous value, but should be statistically distributed and fluctuating with time for each molecule (see Figure 1). With real gases having varying degrees of interaction or binding, the most probable radius may vary from one gas to another. To maintain consistency with Sackur–Tetrode theory, all calculations in this paper have employed the geometric result of 1.7188 × 10^−7^ cm for *r*_t_ at 298.15 K and 1 atm, or 1.6994 × 10^−7^ cm at the Earth’s global average surface temperature of 288.15 K. However, it may be possible to make experimental determinations of the dynamic structure, allowing for more accurate estimates of the action. In any case, the sensitivity to variations is low, given its logarithmic nature and any errors would only cause a slight displacement in the entropy value. Initially, the absolute value of the translational symmetry factor z_t_ was rarely of importance because, in nearly all cases, differences in the entropy of free energy were taken, or the system is isothermal_._ In such cases, the z_t_ factor disappeared.

As a result, we can write the following concise relationship for translational action *@*_t_ and entropy, given a suitable choice of action radius and by incorporating the translational correction factor (z_t_).
*S_t_ = R*ln[*e*^5/2^(*@*_t_*/ħ*)*^3^*] = 2.5*R* + 3*R*ln(*@*_t_*/ħ*)(16)

### 2.2. Rotational Action and Entropy

The Sackur–Tetrode equation was developed for monatomic gases, from which the relationship between action and entropy of this paper was developed. However, it was soon found that the approach was valid for polyatomic gases and could be applied to molecular rotation and vibration. From statistical mechanics (see Moore [6], the rotational contribution to the molar entropy of a diatomic or linear molecule with two-dimensional inertia is given by
*S*_r_ = *R* + *R*ln(8π^2^*kTI*_r_/σ_r_*h*^2^), or *R*ln[e(8π^2^*kTI*_r_/σ_r_*h*^2^)] = *R*ln[e(2*kTI*_r_/*ħ*)^2^/σ_r_].(17)

Equation (8) is two-dimensional only, because there is no significant inertia around the longitudinal axis of a diatomic or linear molecule like N_2_ or CO_2_. The rotational partition function is 8π^2^*kTI*_r_/h^2^; clearly, this can also be recast as an action ratio. Here, the moment of inertia *I*_r_ is given by (*m*_1_*m*_2_/(*m*_1_+*m*_2_))*r*_r_^2^ and *r*_r_ is the average bond length, σ_r_ is the rotational resonance symmetry number (e.g., σ_r_ = 2 for O_2_ and σ_r_ = 1 for NO). In this equation, we can recognise that the rotational action of a gas molecule, @_r_, is equal to (2*kTI*)^1/2^—derived from the rotational energy equal to ½*mr*_r_^2^*ω*_r_^2^ or *I*_r_*ω*_r_^2^/2. Hence, *I*_r_*ω*_r_^2^ equals 2*kT* and *I*_r_*ω*_r_ is given by (2*kTI*_r_)^1/2^, and is equal to @_r_, by definition. Therefore, using similar notation as for translational action and entropy, we have
*S*_r_ = *R*ln[e(@_r_/*ħ*)^2^/σ_r_] = *R* + *R*ln[(@_r_/*ħ*)^2^/σ_r_].(18)

Thus, just as in the case of translational entropy, the rotational entropy can also be expressed as a variable of action alone, given that the symmetry factor σ_r_ is a constant. It is shown elsewhere that action is a function of volume and temperature [3], but volume or pressure changes have little or no effect on rotational entropy as long as the temperature is not too high. All other terms in this equation are constant for a given gas molecule. 

For non-linear gas molecules with more than two atoms, such as the greenhouse gases other than linear carbon dioxide and nitrous oxide, the rotational entropy, *S*_r_, is given from statistical mechanics as
*S*_r_ = *R*ln[{8π^2^(8π^3^*I*_A_*I*_B_*I*_C_)^1/2^(*kT*)^3/2^}/σ_r_*h*^3^] + 3/2*R*,(19)
where *I*_A_, *I*_B_, and *I*_C_ in Equation (19) correspond to the three principal moments of rotational inertia with respect to three perpendicular axes (see Glasstone [5]). In terms of action ratios analogous to those used above, we rearrange this equation to read
*S*_r_ = *R*ln[π^1/2^{(8π^2^*kTI*_A_/*h*^2^)^1/2^(8π^2^*kTI*_B_/*h*^2^)^1/2^ (8π^2^*kTI*_C_/*h*^2^)^1/2^}/σ_r_] + 3/2*R*.(20)

Recalling our previous definition of rotational action *@_r_* of a diatomic molecule as (*2kTI_r_*)^1/2^ for each inertial axis of a linear molecule with more than one atom, we can express the rotational entropy contribution of a non-linear molecule as
*S*_r_ = *R*ln[{π^1/2^(@_A_/*ħ*)(@_B_/*ħ*)(@_C_/*ħ*)}/σ_r_] + 3/2*R* = *R*ln[{π^1/2^e^3/2^(@_A_@ _B_@_C_/*ħ*^3^)}/σ_r_],(21)
where @_A_, @_B_, and @_C_ are the three principal rotational actions for non-linear molecules. Hence, once again, it is possible to express changes in entropy as a simple function of action alone, as all other terms in the equation are constant for a given gas molecule. Given that the product of entropy and temperature *ST* indicates the thermal energy required, there is obviously an exact logarithmic relationship between the total energy required to sustain a system of molecules at a given temperature and the action of each mode of rotation.

The rotational symmetry number σ_r_ for polyatomic molecules depends on the point group of the molecule as defined by the Nobel laureate Herzberg [12]: “A possible combination of symmetry operations that leaves at least one point unchanged is called a point group”. This is a term derived from crystallography, and the characteristic symmetry number σ of each point group can be shown to be equal to “the number of indistinguishable positions into which the molecule can be turned by simple rigid rotations”. Table 1, adapted from Herzberg [13], gives the symmetry number for the more important point groups. Note that methane in the *T* point group has a rotational symmetry of 12, indicating how its quantum field indicated by its rotational entropy is economical for energy in view of its indistinguishable structure regarding its orientation in space. This situation for methane can be contrasted with a similar tetrahedral carbon molecule having only one hydrogen atom in its structure together with three different halogens such as fluorine, chlorine, and bromine. In this case, the symmetry is unity (1.0), such that the energy field has a 12-fold lower frequency of encountering an identical structure in action space.

Quite naturally, this 3-dimensional relationship—which is related to chemical potential rather than kinetics, as we will see later—must be exponential. Hence, more energy per molecule is required to convey information in a molecular field of greater probe-ability, with greater radial separation of identical molecules.

### 2.3. Vibrational Action and Entropy

Vibration in molecules between their atoms occurs at a rate usually much more frequent than rotation, clearly adding to the action of molecules by increasing the length of the trajectory of their atoms in a given time interval. Vibration is characterized by its frequency, which does not change appreciably as more energy is added to a system. On the contrary, the amplitude of vibration increases, affecting the mean kinetic and potential energies and the inertial trajectory of the molecular angular motion. In collision processes, a higher non-equilibrium vibrational energy state resulting from absorption of a quantum of energy in the infrared will act to equilibrate with its rotational and translational energies in the microwave and radiowave bands, effectively dissipating vibrational energy into these modes. This concept will be revisited later when we consider greenhouse gases and how they can affect the gravitational distribution of the atmosphere.

Related to the point groups and numbers of atoms in the gas molecule are the number of vibrational modes; also, the question arises of how many of these modes are degenerate, that is some different vibrations have identical frequencies because of similarities in molecular structure. For a non-linear molecule the number of vibrational modes is equal to 3n−6, where n is the number of atoms. For a linear molecule, the number of possible modes is one less or 3n−5. In either case, 3n is the total degrees of freedom for motion. The number 6 in the case of non-linear molecules is the sum of 3 translational and 3 rotational degrees of freedom. The number 5 in the case of linear molecules refers to 3 translational and 2 rotational degrees of freedom. 

For both non-degenerate and degenerate modes, the vibrational entropy S_vi_ is given by Glasstone [5] as
*S*_vi_ = *R*x/(e^x^ − 1) − *R*ln(1 − e^−x^), where *x* = *h*ν_i_/*kT*.(22)

Here, ν_i_ is the wave number, which equals the number of vibrations per second divided by the velocity of light in cm per second. It therefore has the physical dimensions of cm^−1^. Ultimately, the total contribution to the vibrational entropy is the sum of all vibrations, taking into account any degeneracy where more than one mode of vibration has the same frequency. Then, Equation (22) is derived as follows. According to Moore [6], the vibrational energy *E* is given as
*E = RT*^2^∂lnQ_vib_/∂*T* = L*h*ν/2 + L*h*νe^−*h*ν/*kT*^*/*(1 − e^−*h*ν/*kT*^).(23)

Here, L*h*ν/2 is the zero-point vibrational energy *E*_o_ remaining at absolute zero Kelvin, where L is Avogadro’s number for the number of molecules in a mole. Thus, taking *h*ν/*kT* as equal to x, as used above,
(*E* − *E*_o_)/*T* = *Rx*e^−*x*^/(1 − e^−*x*^) (*A*_vib_ − *E*_o_)/*T* = *R*ln(1 − e^−*x*^) since *A_vib_* = −*kT*lnQ_vib_ = *G*_vib_(24)

Here, *A* and *G* refer to Helmholtz and Gibbs energies.
*S*_vib_ = (*E* − *G*)/*T* = *Rx*e^−*x*^/(1 − e^−*x*^) − *R*ln(1 − e^−*x*^)(25)

Furthermore, the vibrational heat capacity is given as
∂*E*/∂*T = C*_vib_ = *Rx^2^*e^−*x*^/(1 − e^−*x*^)^2^ = *Rx^2^*/(e*^x^* + e^−*x*^ − 2) = *Rx^2^*/2(cosh*x* − 1), given (e*^x^* + e^−*x*^)/2 = cosh*x*(26)

At moderate temperatures, the vibrational entropic energy *S*_vib_*T* contributes mainly as the enthalpy calculated with *C*_vib_, giving both kinetic and potential vibrational energy but with a relatively smaller negative Gibbs energy. It is only at elevated temperatures—when the ratio *h*ν/*kT* is less than 1.0 and as *C*_vib_ approaches 2.0—that *−G,* reflecting positive change in the ‘sum of states’, exceeds *C*_vib_*T*. Then, the statistical Gibbs energy of higher quantum states emerges as more dominant, so that heat is consumed doing quantum work without raising the temperature. 

Despite this, it is clear that each vibration of frequency ν*_i_* contributes its own entropy.
*S*_ν_ = Σ*S*ν_i_(27)

An alternative approach is one with emphasis on the relationship of vibrational quantum states to their translational action.

The probability (P) of a quantum microstate is given in Equation (28).
*P* (microstate *r*) = e^−*Er/kT*^/Σ_i_e^−*Er/kT*^ = *N*_r_/Σ_i_*N = N*_r_/*N*(28)

Hence, the likelihood of a particle being in microstate *r* is given by the number density *N*_r_ divided by the total number density of all microstates. This implies that the ratio of the probable number of particles in any two microstates is (neglecting degeneracy)
*N_r_*/*N_s_* = e^−*Er*/*kT*^/e^−*Es*/*kT*^ = e^−δ*E*/*kT*^.(29)

The work of lifting the molecule from microstate *r* to microstate *s* at the same temperature is given as follows.
−δ*E* = −*kT*ln(*N*_r_/*N_s_*) = −*kT*ln(*r_s_*/*r_r_*)^3^ = −*kT*ln(*@_s_*/*@_r_*)^3^(30)

Thus, the higher vibrational energy of an elevated quantum microstate is matched by a similar elevation in the translational Gibbs energy of the vibrationally activated molecule, as given in Equations (29) and (30). At a given temperature, an orderly progression of vibrational quantum states of exponentially increasing radii occurs, shown in Results. The greater inertial radius of these activated states and their ability to cause chemically reactive conditions is of particular interest for some molecules with weaker bonding. In principle, the greater space-filling property of higher action states with greater radius must have some feature implying added potency. This can be considered as the effective linearity of molecule’s motion and its efficiency in transferring momentum in collisions. The higher the energy state, the greater the action radius and the more effective the impulse because of its greater inertia caused by its higher energised amplitude, reflecting their lower frequency.

It is often possible to measure the energy differences between the different quantum states spectrally, and from this, derive the relative number density of the different states as exponential functions of the translational entropy—known as the Boltzmann or canonical distribution [14]. This form of distribution applies to all variations in quantum states that become entangled, whether translational, rotational, or vibrational. At ambient temperature in the atmosphere, most of the variation in quantum state between molecules is translational, with rotational also varying but less frequently. Elevated vibrational states are even less frequent, reflecting the relative stability of bonds in many organic molecules. Energy differences in translational states are very small, so we find an almost continuous distribution of energy states established around a mean velocity as expressed by the Maxwell–Boltzmann equation.

A flow diagram for computing total entropy and Gibbs energy is given in Figure 2.

## 3. Results

### 3.1. Direct Computation of Molar Action and Entropies from Physical Properties of Atmospheric Gases

We have established several concise relationships expressing entropy as a logarithmic function of an action ratio for translation, rotation, and vibration, using overlays of equivalent excited translational states.
*S*_t_ = *R*ln[e^5/2^(@_t_/*ħ*)^3^]  (translation)(31)
*S*_r_ = *R*ln[e(@_r_/*ħ*)^2^/σ_r_]  (rotation—diatomic or linear molecule)(32)
*S*_r_ = *R*ln[π^1/2^e^3/2^(@_A_@ _B_@_C_/ *ħ*^3^)/σ_r_] (rotation—polyatomic molecule)(33)
*S*_vi_ = *R*x/(e − 1) − *R*ln(1 − e^−*x*^), where *x* = *hν*/*kT* (for each vibrational mode)(34)

It is clear that the translational action ratio @_t_/*ħ* will vary as a function of temperature affecting velocity, but also with volume. Thus, action acts as a surrogate for the effects of both temperature and volume or density for translational entropy. Normally, these variables are considered separately. At extremely low temperatures near absolute zero, the action ratio will tend to a minimum and the entropy will tend to zero, as required by the third law of thermodynamics. Near zero, only vibrational energy remains significant, expressed as the zero-point vibrational energy of *h*ν/2 per bond, proposed as essential by Planck and Einstein [5,6]. In Figure 3, the variation in vibrational energy above the ground state at absolute zero that contributes to molecular entropy is shown for the first three energised modes of carbon dioxide of wavelength 667 cm^−1^. The radii of each state in action phase space exponentially increase. Given that action (*mrv*) at a given temperature of constant mean momentum (*mv*) is proportional to radius, the ratios of the probable volumes from the number density (*N*_n_) can be expressed one-dimensionally in terms of the inertial radius *r* and then to the relative action compared to the ground state.

In Figure 3, the translational action of successive vibrational quantum states and the equivalent translational energy difference (3*kT*ln(@_n_/@_o_)) is computed for the 667 cm^−1^ infrared resonance of CO_2_; similar calculations can be performed for the other infrared resonances at shorter wavelengths, or for those of water molecules. Each successive state of increasing energy occurs with decreasing frequency and, therefore, has a declining number density and increasing inertial radius and action ratio. Trajectories of increasing radius may therefore have greater impact in collisions corresponding to greater translational action. Then, the entropy for the activated states can be computed from the equation
*S*_vi_ = *R*x/(e*^x^* − 1) − *R*ln(1 − e^−*x*^).(35)

Figure 3 confirms the co-variation of vibrational entropy with action, accepting that this equation for entropy contains terms related to both the kinetic energy and enthalpy, as well as for configurational symmetry. Causally, this must be based on the need for specific quantities of latent field energy to sustain action at a statistically stationary value, as appropriate for a particular kinetic environment and temperature. The statistical nature of entropy implicit in Boltzmann’s and Gibbs’ theories must also correspond with the relationship with action, discussed in Kennedy [3].

In Table 2, Table 3 and Table 4, calculations using these equations are indicated for individual contributions to the molar entropy of some common atmospheric and greenhouse gases. The computed entropy values at standard temperature and pressure compare very well with the rounded standard values [15]) obtained experimentally according to the third law of thermodynamics. This statistical correspondence can be found in the space-filling dynamic nature of molecules subject to collisions, so that any complexion of high frequency or pressure involves chemical species occupying a comparatively small volume per molecule with low translational action at a given temperature; those complexions of higher molecular entropy occupy a comparatively large volume per molecule with higher translational action. This is also consistent with Shannon’s information version of entropy, considering information as uncertainty and the capacity of a message to surprise [16].

Primary data for Table 2, Table 3a–c were obtained from Herzberg [12,13] or from the National Institute of Science and Technology (www.NIST.gov). All entropy values were calculated as shown in Figure 2.

An interesting feature of the data calculated for Table 2 and Table 3, but rarely mechanistically considered, is the significantly greater entropy related to translational action compared to rotational and vibrational action for all gas molecules at 298 K. The lower the pressure of a particular gas, the greater this discrepancy. Indeed, most of the heat required to raise the temperature of the gas from 0 to 298 Kelvin under standard conditions is devoted to sustaining translational action, with only a small proportion of molecules exhibiting any vibrational action and entropy at all. If we consider that the relative action states @/*ħ* for carbon dioxide at standard temperature and pressure calculated here are 491 for translation, 24 for rotation, and a much lower number for excited vibration, indicating a very low proportion of molecules excited with infrared quanta, it is reasonable to conclude that the size of the quanta associated with rotation and translational action states have correspondingly lower frequencies. This would appear to place them in the microwave and radiowave range of the electromagnetic spectrum—relatively cold or dark energy. 

Thus, although heat is required to generate translational entropy, its actual form could be considered as gravitational work, although for suborbital molecules with quanta of relatively high frequency. If so, we could consider translational action and entropy as indicating the quantity of heat required to reach *T* degrees Kelvin in the gravitational field, including that normally regarded as pressure–volume work.

We must regard the heat that was required to melt and vaporise the carbon dioxide molecules as having performed configurational work, either on separating the molecules or pressure–volume gravitational work of lifting the atmosphere. This work-heat consumption identified by Clausius [1] in 1875 explains the fact that not all the heating included in the entropy function contributes to sensible heat just raising the temperature as increased kinetic energy, consistent with the heat capacity of each molecule. Similar conclusions can be drawn for the melting of ice to water and its subsequent vaporisation. Such interpretations regarding interconversions of heat and work require further investigation, and this may be facilitated using the quantum features of the action approach.

### 3.2. Considering Water’s Phase Changes

Of all the atmospheric gases considered here, only water exists on Earth as gas, liquid, and solid. This erratic cycle is very apt for illustrating significant changes in entropy states associated with changes in phase. Most of the permanent gases in the atmosphere only exist as vapours. As a result, their changes in entropy refer only to changes in kinetic and potential energy corresponding to changes in enthalpy and in free energy as a response to changes in temperature and pressure, respectively. However, water has highly significant changes in action and entropy in the atmospheric weather cycle, with corresponding consumption or release of heat. 

The total thermal capacity to bring a mole of water to vapour at 298.15 K and 1 atmosphere pressure (were this possible) is 56.2 kJ. Of this, only 7.4 kJ can be attributed to its heat capacity as increased kinetic energy per mole over the temperature range, with 5.8 kJ required for melting and 44.0 kJ to vaporisation at 298 K. Almost 90% of the solar heat absorbed by water as vapour in the atmosphere is available for release in the two phase transitions of forming snow or hail. Although these facts are well known and the major warming possible during atmospheric condensation of water vapour is understood, this could also be a fruitful area for further investigation using the action-entropy theory. 

Considering Table 3 and Table 4, it is of interest that water has only minor vibrational entropy at ambient temperatures—lower even than that of oxygen, mainly at very similar short infrared wavelengths (6.270 and 6.329 μm)—potentially allowing overlap for emission or absorption. On this basis, there might seem to be only a weak case to consider water a major greenhouse gas, although this is customary. In fact, the converse is true, as water’s low actual excitation of vibration to a higher quantum state—particularly by the two shorter wavelengths for water—results in a high population of water molecules remaining in the ground state, able to be excited by radiation from the Earth’s surface. However, the reversible latent heat of vaporisation of water, released as infrared quanta around 6 μm when it condenses in clouds or at dewpoint [17], is also an important factor for radiative heat transfer in the atmosphere. 

It is also apparent that the actual vibrational entropy of methane is only about one-sixth that exhibited by carbon dioxide and nitrous oxide (Table 4). Presumably, the low vibrational entropy can also be related to a higher residual absorptivity of methane, but it is only slightly statistically enhanced by being poorly excited at this temperature. Only a very low proportion of methane, as well as water molecules, are excited by infrared radiation at equilibrium under ambient temperature conditions, as shown by their low vibrational entropies at 298.15 K.

As shown in Table 2 and Table 3, the organohalogens that have been withdrawn from use for environmental reasons, such as Freon-11 (CFCl_3_), have an exceptionally large vibrational contribution to entropy. Replacing the hydrogen atoms of methane with these two halogen atoms also significantly increases both the rotational and vibrational entropy; this should lessen absorptivity in the longer infrared region significantly, since this is relatively excited at 298 K as a result of longer bond lengths and greater ease of dissociation of atoms. According to Glasstone [5], the formula for calculating vibrational entropy strictly applies only to divalent molecules. However, this cautionary note may not be required for polyatomic molecules after all. By applying the formula to each bond separately and summating as shown in Table 3b, including any degeneracy, the agreement with experimentally determined entropies using the third law approach is just as good as for other molecules where only translational and rotational contributions are significant. 

Nitric oxide (NO), although not a greenhouse gas with only one vibrational line in the shortwave infrared (5.252 μm), is included for comparison, as are CO (4.608 μm) and O_2_ (6.329 μm) (Table 2 and Table 4). For nitric oxide (NO), a large discrepancy in total entropy between the data calculated here from translation, rotation, and vibration would occur if the electronic (Q_e_ = 4) term was neglected, as a result of its free radical nature containing unpaired electrons; these add *R*ln2^2^ or 11.53 extra entropy units per mole, giving a total value of 211.1, in agreement with the Aylward and Findlay value [15].

The Sackur–Tetrode equation includes a term for the electronic partition function (Q_e_). In the case of O_2_, the ground state electronic partition function Q_e_ is 3 at STP because this molecule has two unpaired electrons that can have their two spins oriented three ways with respect to the nuclear spin —both up, both down, and oppositely. Since they can be distinguished, the three different oxygen species have three times the volume per particle, affecting their action because of the greater radial separation than if only a single species existed. This gives an additional electronic entropy contribution of *S*_e_ = *R*lnQ_e_ or *R*ln3.

However, the fact that a mole of oxygen will contain one-third of a mole of each species must be also be considered in estimating the total entropy. This Q_e_ factor is included in the Sackur–Tetrode equation as *S*_t_ = *R*ln[Q_e_e^5/2^(*V*/*N*)(2π*mkT*)^3/2^/(*h*^3^)], or in the action form of the equation as *R*ln[e^5/2^(@_t_/*ħ*)^3^Q_e_], which is equal to *R*lnQ_e_ + 2.5*R* + 3*R*ln(@_t_/*ħ*). Thus, asymmetry (Q_e_) increases entropy by increasing the spatial distance between molecular interactions, and symmetry decreases it by reducing spatial distances since less field energy is needed to sustain a symmetrical molecule than an asymmetrical one.

Due to the statistical variation in quanta for rotational and translational fine structure, the actual vibrational spectra are not confined to these spectral lines but distributed around these wavelengths. The spectra may be sharpened by cooling the gas, and this is usually done when testing the theory with data. 

The data in Table 2 and Table 3 were calculated for standard conditions of temperature (298.15 K) and pressure (1 bar or 10^5^ Pa^)^. To adjust these results to the actual gas pressures in the atmosphere at the same temperature, only the translational action and entropy will vary. In terms of centimetre–gram–second (cgs) units, the pressure is equal to *kTa*^−3^ or *kT/*8*r*^3^ at 1.013 × 10^5^ pascals, being the product of the mass of air per square cm of the Earth’s surface (ca. 1 kg) and the acceleration of gravity (9.807 m∙s^−2^). 

It is suggested here that entropy can be considered a dimensionless number, corresponding to its role in probability, since it expresses a total thermal capacity per degree of temperature (J/°C)—a ratio of extensive and intensive measures of energy, given that *mv*^2^ is equal to 3*kT* and Boltzmann’s constant *k* is also taken as dimensionless. It is also instructive to be aware that the product of entropy and absolute temperature (*ST*) is always a significant multiple of the kinetic energy since that is merely one of its components—the sustaining field energy corresponding to decreases in free energy from absolute zero must be added to its kinetic energy while heating the molecules, absorbing any heat that becomes latent during this process and in doing any work, such as breaking H-bonded aggregated structures or pressure–volume work against the atmosphere. In this connection, much of the magnitude of *ST* is generated together with increased enthalpy during phase changes when parent solid or liquid matter is melting or vaporising. Gibbs energy does not change when these reversible processes occur isothermally. The chemical potential of the liquid water is equilibrated with that of the vapour at the boiling temperature, with the increase in enthalpy on vaporisation being effectively an increase in internal entropies associated with increasing the internal vibration and rotation of the declustered water molecules. Such increases in internal action and entropy are actually increases in enthalpy.

Any decrease in the density of a chemical substance such as the expansion of a solid, liquid, or gas will also increase *ST* as its Gibbs free energy decreases. For example, liquid water gradually changes its state during heating from large H-bonded clusters of about 30 water molecules just above freezing to fewer than half that number per cluster just below boiling temperature [18], above which the clusters completely dissociate. The variable action of these flickering clusters between 0 and 100 °C could be calculated and the changes in entropy estimated. 

Some results calculated for the actual sea level pressures expressed in atmospheres of all atmospheric gases at the standard temperature of 298.15 K are given in Table 5. Entropy values are given per mole of each substance, using Boltzmann’s constant *k* multiplied by Avogadro’s number N as the unit value and the product (entropy × temperature, *ST*) estimated for each as a proportion of the total.

The total entropic energy as Σ*ST* in air at sea level is about 2.4 MJ per cubic metre. It is obvious that the very dilute gases like nitrous oxide and methane have a relatively large translational entropy compared to the major gases and therefore need more heat per molecule to bring them to this temperature and pressure. The majority of the heat required (*ST*) to raise the atmosphere is absorbed into the fields of only three different molecules—nitrogen, oxygen, and water. Given the reversible phase changes available to water, most of its maximum entropic energy is made available during condensation as part of the hydrological cycle. Roderick et al. [19] have estimated that for a warming of 2.8 K, the atmospheric content of water would increase from an equivalent liquid column of 30 to 35.9 mm, or 7% per K of warming. According to Table 5, by proportion alone, this would amount to 4241 J per cubic metre of air at the surface of extra heat required. However, an exact calculation would need to consider the diminution of the translational entropy per molecule as a result of its increased concentration.

In Table 5, this operation is only illustrated for the densest surface layer of the atmosphere. If the temperature and pressure of the gas is known, a similar calculation can easily be repeated at all altitudes in the troposphere or the stratosphere, and the results integrated to give the total heat capacity of the atmosphere, a task that others are invited to perform. For such calculations of action and entropy, it is convenient to use suitable computer programs. A fully annotated outline of such a program is given as supplementary material or on request from the corresponding author.

### 3.3. Phase Space as Action Space

As shown in the equations above, the mean molecular free energy can always be expressed as a variable of the relative action ratio *@/ħ* alone. Entropic energy (*sT*) also includes kinetic energy and the capability for pressure–volume work. This extends to vibrational and electronic states although their contribution to atmospheric gases near the surface of the Earth at ambient temperatures is usually relatively small, as shown in the tables. Using the action model, the negative relationship between free energy and the entropy is more clearly revealed. Paradoxically, “free energy” is not real energy, but actually denotes its absence; it is better viewed as a system’s potential for action or higher quantum states showing its capacity to accept thermal energy by increasing its action and sustaining field energy, building more complex and diverse internal structures whilst doing external work such as expanding the atmosphere or lifting weights in a gravitational field. To the extent that cooling gravitational work that increases the free energy is conducted, heat may re- later emerge if the reverse work is conducted on the molecules of the system. This reversibility is the essence of the second law of thermodynamics in action theory. 

Given that the relative action or mean quantum number *@/ħ* can be expressed simply as a function of the particle’s mass, its radial separation and the square root of the temperature affecting velocity, all of the paradoxes regarding entropy—such as its lack of change during the mixing of equal volumes of identical gases versus the change when two distinguishable gases are mixed at the same final pressure and temperature—are easily resolved. Each of the distinguishable molecules now occupy twice the space as before, increasing their action accordingly, whereas identical gases must remain in the same space with the same action as before mixing.

In principle, the suggestion to calculate entropy from the logarithm of the translational and rotational action (including its modification by vibration) is far from new. Gibbs identified the significance of action in his classical text of 1902 [8], describing it as the extension in phase (*V*_pq_), claiming that “the quantity … which corresponds to entropy is log *V*, the quantity *V* (not volume) being defined as the extension in phase”. Hence, we can conclude that according to Gibbs, even before Planck identified his quantum of action, any equipotential contour in phase space of equal translational action (*V*_p_ × *V*_q_ = *mv* × *r*) would also correspond to states of equal translational entropy. In effect, changes in the momentum *mv* and a linear coordinate *r* would lead to no change in their product action and its logarithm, entropy. We can now recognise such contours as adiabatic, differing by a minimum of Planck’s quantum of action *h*, giving a scale for estimating maximum uncertainty in momentum or position. Hence, this paper can be considered as a 21st century quantum revision of Gibbs’ 19th century suggestion.

No claim is made here that the action model is inherently more accurate than the classical methods of statistical mechanics or that the formulae given here for ideal gases apply without corrections under all conditions of temperature or pressure. However, the results are easily obtained from primary data and are surprisingly accurate, even for vibrational entropy. This suggests that the action method will have strong heuristic value—not only for climate science, but also for theoretical and experimental purposes in all branches of chemistry and physics. For example, this action revision of the nature of entropy and free energy and the interaction between internal and translational action states has the potential to advance the reaction rate theory and many other processes occurring in the liquid state, including those of life systems. In this area, the translational action will play a special role, since it is closely related to changes in Gibbs energy for molecular trajectories from the chemical potential of free reactants through reversibly activated transition states to the chemical potential of products (see Kennedy [3], Chapters 4 and 6). 

### 3.4. Boltzmann’s Realistic Collision Model of Entropy

Strongly relevant to the equations for entropy based on logarithmic functions of action given in this paper is the approach used by Ludwig Boltzmann [7]; he derived an equation for entropy using his *H*-theorem by considering the mathematical behaviour of a collision integral using a realistic model. This theorem was based on integrating the average effect on a single molecule of collisions with all the other molecules of the gas, spontaneously increasing its entropy whenever commencing with a more ordered state. This led him to essentially the same equations for entropy as those of Gibbs, while claiming “that the mechanical basis is necessary to illustrate the abstract equations”, despite the current of opinion at the time from Mach and others directed against the existence of molecules. For example, Boltzmann [7] gives an expression for the integral of the sum of the entropies of the masses in the volume elements as [*R*ln(ρ^−1^*T*^3/2^) + const.], where ρ is the number density of gas molecules. We can observe that this result only lacks the quantum of action *h* as a suitable divisor in the logarithmic term to remove the physical dimensions of action per unit mass. 

In general, only differences in action and entropy between alternate states are of thermodynamic interest, since the absolute entropy is not required. For translation, we have shown that the entropy change per mole, say for elevation in the atmosphere, given a change of state (1=>2) can be expressed as the function shown in Equation (16).

Δ*S*_t2-t1_ = *R*ln[(e^5/2^*@*_t2_*/ħ*)^3^] − *R*ln[e^5/2^(*@*_t1_*/ħ*)^3^] = 3*R*ln[(*@*_t2_*/*(*@*_t1_)](36)

Therefore, the precise choices of the symmetry factor or of the translational radius are only of significance for estimating the absolute entropy. Due to the statistical nature of momentum and position in phase or action space, assigning an exact value to the most probable radius or symmetry factor to each molecule is impossible, since they will fluctuate around statistical mean values. However, the amplitude of the fluctuations is of interest, since these will control rates of transition.

### 3.5. Under Isothermal Conditions Gibbs Energy Varies with Translational Action and Entropy

Given that the third law of thermodynamics states that the entropy at the temperature of absolute zero is 0, this would require that the action ratio *@_t_/ħ* for a gas at this minimum temperature must be slightly less than 1, if it could exist as such, since the translational entropy can then be considered as equal to *5/2R* + *R*ln[(*@*_t_*/ħ*)^3^], equivalent to the Gibbs expression for entropy of *ST = H−G*, where *G* is the free energy or work potential of a monatomic gas at constant pressure. This suggests that the magnitude of the function *RT*ln[(*@*_t_*/ħ*)*^3^*] has the same value as the free energy, although opposite in sign, so that *G* = −*RT*ln[(*@*_t_*/ħ*)*^3^*] or *RT*ln[(*ħ/@*_t_)^3^] and Δ*G* = 3*RT*ln[(*@*_tr_*/*(*@*_tp_)] for changes in action state at constant temperature. 

Incidentally, the total entropy change during an isothermal chemical reaction includes contributions from changes in translational action and entropy at *T* − δ(*S*_t_T) is equal to the change in Gibbs energy—as well as changes in the internal action and entropy representing changes in enthalpy as a result of revised bonding energies. It is important to understand that the enthalpy term designated *H* refers to the sensible heat that tends to change temperature. Thus, if a chemical reaction results in products where atoms or electrons are more firmly bound with shorter radii, the reduced potential energy will be compensated by increased internal kinetic energy and equal quantities of emitted quanta, resulting in a release of heat as a reduction in Gibbs energy and an increase in entropy of the surrounding system. In the absence of such chemical reactions, the enthalpy change can be measured by the changes in kinetic energy and pressure–volume work alone. This is also true with monatomic noble gases like argon. 

It is important to note that, within limits, the internal entropy for rotational and vibrational states is a function of temperature only. It is unaffected by changes in concentration, except at very high densities, in contrast to translational states. A low concentration or pressure corresponds to a high action state of greater entropy. Therefore, at constant temperature, changes in free energy are purely a function of changes in translational action states, since internal entropy or enthalpy remains constant, or fluctuates around a stable mean value though variations in internal states by absorption or emission of radiation and re-equilibration with translational states. This is entirely consistent with chemical work processes being directly mediated by translational inertia and pressure, such as pressure–volume work. 

We can then write that
*G* = *H* − *ST* = *H* – (*S*_t_*T* + *S*_r_*T* + *S*_vi_*T*)(37)

The enthalpy (H) is a term always referring to the sensible heat in a system that can be measured with a thermometer and related to the kinetic energy of its molecules. The entropic energy *ST* differs in that it is only partly indicative of sensible heat, but includes the potential energy stored in work such as thermodynamic work in molecular systems, or gravitational work. According to the Carnot principle, this work can reappear as sensible heat in a reversible system, raising the enthalpy with the temperature. This potential source of extra warming certainly applies to gases in the Earth’s atmosphere. Indeed, it is responsible for much of the heat transfer to higher latitudes, released by frictional processes on the Earth’s surface. 

For monatomic gases, we can rewrite this classic equation taught to all students using the algorithms developed here as follows.
*G* = *H* − *ST* = *− RT*ln[(@_t_/*ħ*)^3^] = 1.5*RT* + *RT* − *RT*ln[e^5/2^(@_t_/*ħ*)^3^](38)

For the main diatomic gases in the atmosphere, nitrogen and oxygen, at ambient temperatures, we will observe the following when including rotational entropy and neglecting vibrational entropy.
*−RT*ln{[(@_t_/*ħ*)^3^Q_e_][(@_r_/*ħ*)^2^/σ_r_]} = 2.5*RT* + *RT* − RTln{[e^7/2^(@_t_/*ħ*)^3^Q_e_] [(@_r_/*ħ*)^2^/σ_r_]}(39)

Disallowing a role for the enthalpy of chemical reactions at ambient temperatures in the troposphere, we have
*G* = *H* − *ST* = 3.5*RT* − *ST*.(40)

Alternatively, we can write a modified equation for Helmholtz energy in constant volume conditions by varying, only slightly, the *RT* or *PV* term.
*−RT*ln{[e(@_t_/*ħ*)^3^Q_e_][(@_r_/*ħ*)^2^/σ_r_]} = 2.5*RT −* RTln{[e^5/2^(@_t_/*ħ*)^3^Q_e_] [(@_r_/*ħ*)^2^/σ_r_]}(41)
*A* = *E* − *ST* = 2.5*RT* − *ST* = *G* − *RT*(42)

For polyatomic molecules or at temperatures where vibrational entropy and energy are more relevant, it is simple to add the vibrational entropy terms to both sides of the equation.

In his engaging book on statistical mechanics, Schrödinger [20] derived the relationship *nk*ln*ζ* = *U* + *PV* − *TS* by calculating from the ‘sum over states’ Σ*N*_i_/*N* = *e^−ε1/kT^* + *e^−ε2/kT^ + e^−ε3/kT^+ e^−ε4/kT^ …+ e^−εn/kT^*; *nkT*ln*ζ* is the thermodynamic potential (or free energy) for *n* molecules, a function of an inverse action ratio *ζ*. He defined the factor 1/*ζ* as a function of the translational partition function (*2πmkT/h^2^*)^3/2^*V*, divided by the number of particles (*n*)—that is, as a translational action ratio as defined in this paper. By contrast, its inverse *ζ* is an ‘inaction’ ratio, indicating the free energy, and Schrödinger’s insightful equation precedes, by at least 70 years, the action potential theory of free energy given here. For a perfect monatomic gas, *PV* is equal to *RT*, and so *U* + *PV* is equal to the enthalpy *H*, which does not change for individual molecules of a chemical species unless the temperature changes.

In contrast to translation and rotation, vibrational action states higher than the ground state are largely unoccupied at ambient temperatures, and most greenhouse molecules in the atmosphere are still in their coldest vibrational states, despite them radiating as required by the temperature, but falling away by the fourth power of the temperature in Kelvin. Thus, vibrational action and entropy are minimal. This ordered state of low vibrational entropy is fortunate for life on Earth, otherwise stable molecules and structures would be impossible. In kinetic theory, it has usually been assumed that molecular trajectories are linear, with no interaction between molecules. 

Whether the translational trajectory of the molecules is considered as curved or straight is irrelevant, given that the speed of energy transfer vastly exceeds that of the molecules; relatively to the speed of transmission of the thermal field energy bath referred to by Clausius in 1875, molecules are almost stationary. 

### 3.6. Greenhouse Gases and Temperature Equilibration in the Gravitational Field

When individual molecules are heated internally by absorption of infrared radiation, increasing their vibrational action and entropy, the rotational and translational modes of action will respond almost immediately, mediated through subsequent collisions occurring within microseconds [21]. The absorption of quanta will decrease their internal free energy whilst increasing their inertia and capacity to exert pressure, potentially doing gravitational work while moving to higher altitude and thus lowering the local temperature as kinetic energy declines. This may seem paradoxical, but it is consistent with the virial theorem [3], and this idea was developed further in a companion paper [22] assessing the possible role of greenhouse gases on temperature gradients with altitude. 

Such dissipation processes for absorption and emission of radiant energy may give a special role for greenhouse gases in the atmosphere, since the major gases nitrogen and oxygen have little, if any, such absorptive activity. Their presence enhances the rate of transfer of radiant energy from the Earth’s surface to non-absorptive molecules at higher altitudes. Indeed, this is an important role of polyatomic gases like water and carbon dioxide. On the whole, greenhouse gases are regarded negatively because of their proposed role as agents in global warming; but it is important to also consider their possible benefits for experimental testing, such as enabling elevation of the atmosphere and cooling the surface of the Earth. Once heated, gases with higher heat capacities (including nitrogen and oxygen) also tend to cause the atmosphere to be more elevated because the natural temperature lapse rate with altitude is less than for monatomic gases of similar mass. Thus, an atmosphere of carbon dioxide of mass 44 daltons would be more elevated than one of argon of mass 40, despite its greater weight.

We can examine the relative absorptivity of the greenhouse gases and the existence of absorptive/emissive lines in the infrared (Table 2 and Table 3), recalling that the Earth’s surface has a maximum emission range of around 5–30 μm wavelength (10,000 cm^−1^ = 1 μm, 1000 cm^−1^ = 10 μm; 100 cm^−1^ = 100 μm) whereas sunlight is confined to the 0.3–5 μm range. The longer wavelength of terrestrial radiation compared to sunlight is a result of absorption of sunlight by surface materials, and the re-equilibration of quanta with the much cooler surface of the Earth, compared to the boiling ocean of hydrogen atoms of the Sun. Obviously, polyatomic molecules absorb in the 5–30 μm wavelength band of the infrared, and the more complex the molecules are, the greater the number of absorptions. 

The quanta associated with changes in rotational and translational action must be of longer wavelength in the microwave and radiowave range of frequencies not resonant with the Earth’s major energy primary emissions from sunlight. However, the infrared radiation absorbed by greenhouse molecules will be converted to these lower frequencies as a result of work done in subsequent molecular collisions during temperature equilibration in the atmosphere—the process is known as equipartition. It is of interest that the quanta able to promote equality of kinetic temperature with equilibrated molecules range from infrared for vibrational freedom to microwaves for rotation and radiowaves for translation, thus broadening the spectrum of the energy involved. Changes in such radiation fields during the dissipation of such radiation from Earth should already be detectable using suitable spectrometers on satellite systems.

According to Clausius [1] and the second law, the atmosphere would need to be hotter than the surface to heat the Earth’s surface as a net process. Consistent with this principle, most of the temperature increase at the surface of the Earth from energy fed back from the atmosphere must be a result of the reversal of convective processes in high-pressure zones when air is descending. The fall of atmospheric gases from higher gravitational energy is a work process generating heat, with all air molecules simultaneously gaining kinetic energy and radiating equivalent heat quanta as required by the virial theorem of Clausius [11]. Clearly, this process can heat the surface, as occurs in high-pressure zones or anticyclones. However, this transfer of heat from the atmosphere must be balanced by compensating transfers of radiant heat into the atmosphere in low pressure zones as gravitational work is performed using heat. These reversible processes demonstrate the Carnot principle that so impressed Clausius. Surprisingly, in climate science, little attention is paid to the reversible transfers between heat and work that are implied in Lagrange’s earlier identity relating the second derivative of the inertia of a system of particles (*I* = Σ*mr*^2^) with respect to time and its kinetic (*T*) and potential energy (*V*); these can be considered as surrogates for heat and work in a gravitational or central force system.

½d^2^*I*/d*t*^2^ = 2*T* + *V*(43)

On this basis, inertial effects can be considered as sources of heating or cooling, as seen in convection and advection near the Earth’s surface (Kennedy [2], Chapter 5). In fact, this equation should be considered as the basis for the whole of climatology, a contention we begin to explore elsewhere [11,21]. A simplifying aspect of the virial theorem for conservative systems is that changes in action state involve equality of variations in total energy and kinetic energy though opposite in sign. Thus, the absorption or the emission of energy quanta as changes of state require instantaneously equal decreases or increases in kinetic energy, respectively. This is the basis for the conclusion that variations in potential energy shown in equation (21) are twice the magnitude of variations in kinetic energy. 

Table 3 and Table 4 show the specific frequencies of infrared radiation from the Earth that different gases such as CO_2_, N_2_O, and CH_4_ will absorb. However, a CO_2_ molecule activated by IR absorption to vibrate more vigorously will transfer most of this energy to other air molecules in the next collisions, thus increasing their action and entropy while dissipating the activated internal state and increasing their Gibbs energy. Furthermore, the more dilute the gas (e.g., N_2_O and CH_4_), the greater its translational entropy, although its vibrational and rotational entropies will be purely a function of temperature. Thus, on absorbing a specific quantum of IR radiation (exciting molecular vibration), such a dilute gas will have a larger disequilibrium between its vibrational action and its translational action. In a subsequent collision, the greater inertia and amplitude of the vibrating atom should cause a more efficient transfer of momentum to surrounding air molecules, irrespective of whether they are greenhouse gases or not.

In the action theory, emphasis is placed on the fact that the greenhouse gases act to capture the Earth’s radiant energy and the momentum associated with these quanta; the greenhouse gases are then accelerated in their vibrational, rotational, and translational energy as a result, all tending to increase the temperature and all atmospheric molecules responding with increased action and entropy. Since the quanta from the Earth are directed towards outer space, there is even a small radiant force (Σ*hν_i_/cδt*) tending to selectively elevate the greenhouse gases, compared to the non-absorbing gases, N_2_ (78%), O_2_ (21%), and Ar (1%), but the thermodynamic action potential to elevate greenhouse gases outlined here—as a result of the opposite function of internal and external translational action and entropy—is much greater. 

Hence, this thermodynamic force and heating effect is transferred to N_2_ and O_2_ as a result of collisions, and the heated gases expand to higher altitude, exchanging their increased kinetic energy for increased gravitational energy and cooling as a result. Perhaps it is more apt to consider that the greenhouse gases, such as water, play an important role in holding up the sky, enabling reversible gravitational work, thereby cooling the atmosphere. These adjustments of temperature of the troposphere allow the outgoing longwave radiation to balance the incoming solar shortwave radiation, providing temperature equilibrium. The average temperature at the surface is automatically adjusted to ensure this balance, fluctuating according the rate of heat flow from the Earth. 

### 3.7. Adiabatic Processes

Adiabatic changes may occur at constant gravitational potential, as when a parcel of air moves laterally by advection, doing electrical work on a wind farm. When a parcel of air moves reversibly by adiabatic convection to a higher gravitational potential, we have to consider the cooling effect of doing gravitational work in addition to changes in the atmospheric pressure. The lower the pressure exerted by the weight of the atmosphere above the parcel of air, the less pressure–volume work and heat that is needed for expansion. However, almost the same amount of heat is required to raise the gravitational potential of the same parcel of air regardless of the altitude. 

By contrast, a descending parcel of air may be adiabatically compressed and spontaneously heats as gravitational potential declines causing kinetic work and internal heat-work varying free energy to be done on the air as it falls. We will show elsewhere that the increase in kinetic heat shown by the temperature increase at the expense of gravitational potential energy is matched by the decrease in free energy of the thermodynamic field, also consistent with the virial theorem. Furthermore, the capacity to do work of the air parcel declines as the atmospheric pressure increases and pressure–volume work becomes more costly. 

As appropriate for statistical thermodynamics, these calculations of entropy and free energy relate only to the scale of randomized molecular motions of canonical ensembles. Neither the kinetic energy nor the “work-heat” or potential energy involved in convective and advective motions of parcels of air has been considered here. The thermal energy required to initiate these higher order motions (i.e., neither vibrational, rotational, nor translational) is substantial, even though the kinetic energy generated is relatively minor compared to that of the randomized molecular motions. However, the “work-heat” required for anticyclones and cyclones generated by thermal gradients in the gravitational field is substantial. The observation, here—that the major part of the heat required per molecule (*sT*) from absolute zero to 298 K relates to the latent or “work-heat” compared to the sensible kinetic heat—is a striking observation that is rarely made. For example, for argon, the total entropy (18.6*k* per molecule or 154 J/C/mole) shown in Table 3 at 1 atm, is 12.4 times the increase in entropy from 0 K for kinetic motion alone (1.5*k*). At 0.01 atm in the atmosphere, the ratio for configurational energy is even greater. For all the molecules in a rotating parcel of air, the potential energy or “work-heat” of motion in these coherent “fly-wheels” is orders of magnitude greater than the kinetic energy of their circulation. Even though the dissipation of this “work-heat” as frictional heat at higher latitudes is a major mechanism for the dispersal of solar energy from the equator towards the poles, this source of warming is rarely properly considered in climate models. This should change.

## 4. Conclusions

Unfortunately, in recent years, thermodynamics and statistical mechanics have largely fallen into disuse, except by specialists. However, this need not continue, given the ease of calculating accurate values of entropy and free energy of gases displayed in this paper. We intend to extend this simple link—between theory and practice—obtained with gases to the liquid state and the vibrations of macromolecules. Furthermore, interpreting thermodynamics in terms of physical action states provides a realistic modelling approach that can simplify the study of the links between heat, work, and morphogenesis [2]. Linking vibrational quantum states to translational action states as shown in Figure 3 provides a new dimension for establishment of reactive morphology. The conclusion that higher vibrational energy states have equivalent expressions as translational action and energy of provides a direct connection between quantum states and space-filling property. We have commenced applying this action approach to modelling the troposphere [11,22] also applying the virial theorem to generating temperature gradients with altitude, yielding a steady state lapse rate between −6.5 and −6.9 °C per km, depending on the water vapour content, with a profile of increasing entropy vertically. We recommend widespread application of this explanatory approach.

## Figures and Tables

**Figure 1 entropy-21-00454-f001:**
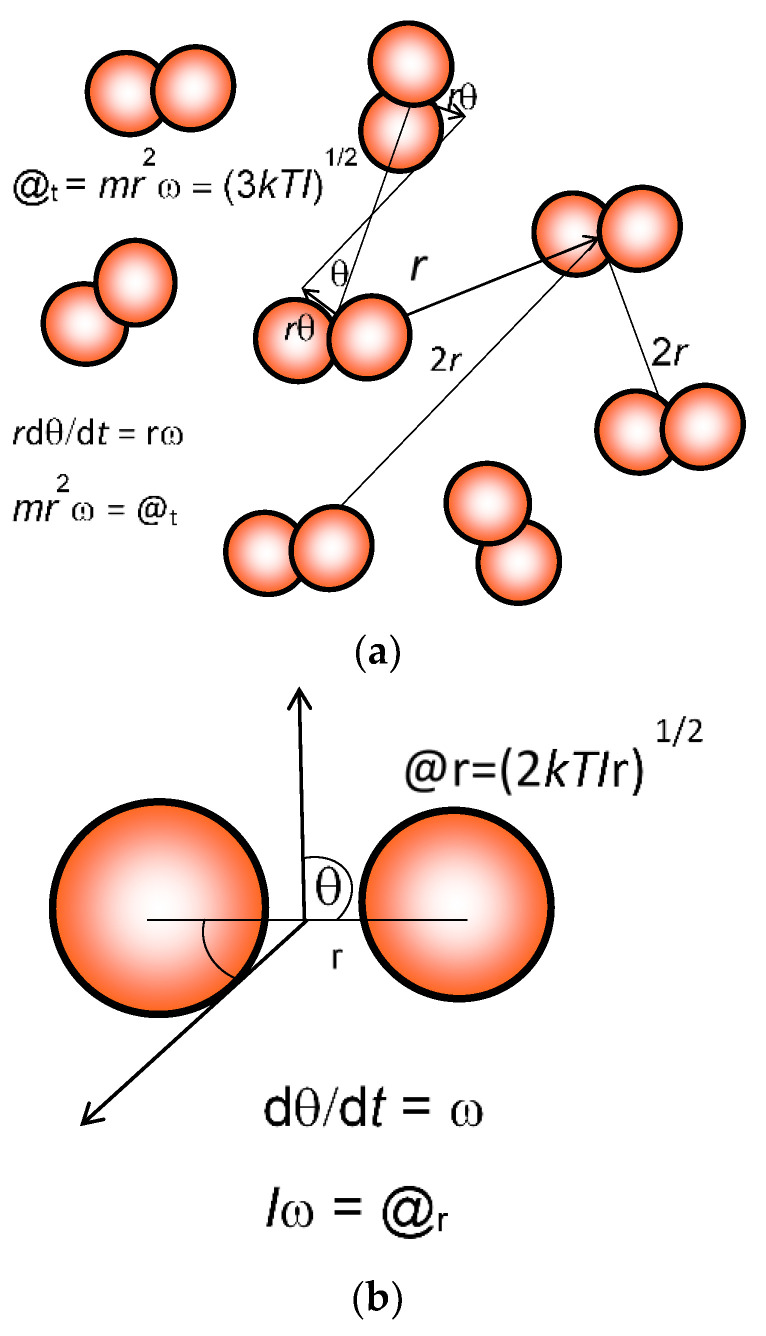
Calculation of translational and rotational action (@). Mean translational action @_t_ (**a**) is estimated, as explained in the text, from the average separation of *a* = 2*r* by allocating each molecule a space of *a*^3^ = *V*/N, where *V* is the total volume and N is the total number of diatomic molecules like dinitrogen (N_2_). Relative angular motion dӨ/d*t* = ω is estimated for molecules exhibiting the root mean square velocity, taking 3*kT* = *mv*^2^ = *mr*^2^ω^2^. Then translational action @_t_ is equal to [(3*kTI*_t_)^1/2^/2.170806]. Rotational action @_r_ (**b**) for linear molecules such as N_2_, O_2_, and CO_2_ is similarly estimated, and equated to (2*kTI*_r_)^1/2^. *I* is the moment of inertia equal to *mr*^2^.

**Figure 2 entropy-21-00454-f002:**
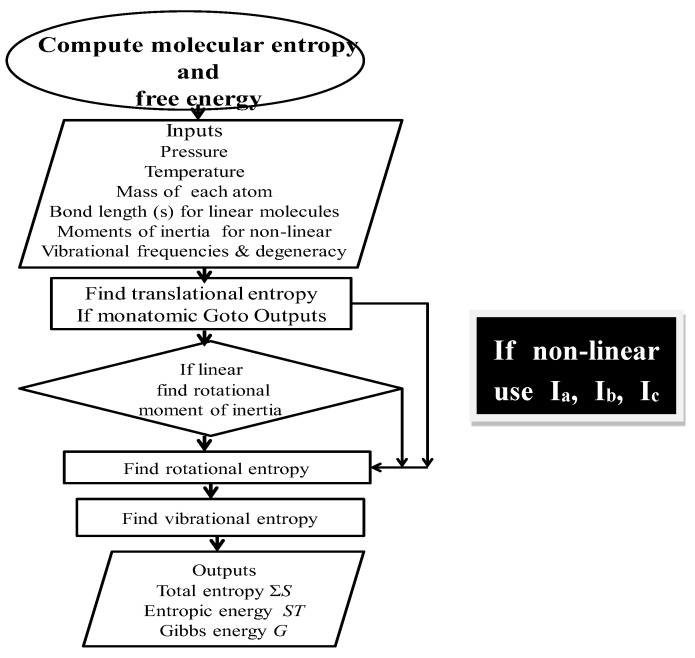
Flow diagram for computing absolute entropy and Gibbs energy. I_a_, I_b_, and I_c_ refer to the inertial moments of inertia. A fully annotated description of the relevant algorithms and subroutines to compute entropy and free energy is available online at the Entropy site, or on request to the corresponding author.

**Figure 3 entropy-21-00454-f003:**
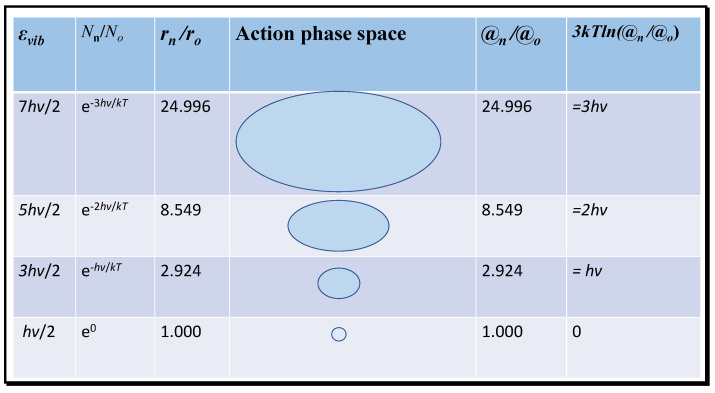
Translational action of activated states for the weak 667 cm^−1^ vibration of CO_2_.

**Table 1 entropy-21-00454-t001:** Symmetry numbers for various point groups. (Modified from Herzberg [13] p. 508).

Point Group	Symmetry No.	Point Group	Symmetry No.	Point Group	Symmetry No.
	σ_r_		σ_r_		σ_r_
*C_1_*, *C_i_*, *C_s_*	1	*D_2_*, *D_2d_*, *D_2h_*	4	*C_∞_*	1
*C_2_*, *C_2v_*, *C_2h_*	2	*D_3_*, *D_3d_*, *D_3h_*	6	*D_∞h_*	2
*C_3_*, *C_3v_*, *C_3h_*	3	*D_4_*, *D_4d_*, *D_4h_*	8	*T*, *T_d_*	12
*C_4_*, *C_4v_*, *C_4h_*	4	*D_6_*, *D_6d_*, *D_6h_*	12	*O_h_*	24
*C_6_*, *C_6v_*, *C_6h_*	6	*S_6_*	3		

**Table entropy-21-00454-t002a:** (**a**)

Gas	MW	I_t_ × 10^40^ (g·cm^2^)	@_t_/ħ = n_t_	S_t_ = Rln[e^5/2^(n_t_)^3^] (J·K^−1^)	Radius × 10^10^ (cm)	I_r_ × 10^40^ (g·cm^2^)	@_r_/ħ = n_r_	S_r_ = Rln[e(n_r_)^2^/σ] (J·K^−1^)	1/λ cm^−1^	x = h ν/kT	S_v_	Q_e_	S_e_	ΣS
H_2_	2	98.83	104.757	117.48	74	0.458	1.8413	12.7	-	-	-	1	0	130.18
N_2_	28.013	139.02	392.901	150.45	110	14.235	10.2654	41.27	-	-	-	1	0	191.73
O_2_	31.999	158.12	419.02	152.06	121	19.59	12.0426	43.93	1580	7.63	0.04	3	9.13	205.16
CO	28.011	138.33	391.923	150.39	113	14.643	10.4115	47.27	2170	10.47	0	1	0	197.67
NO	30.006	148.54	406.134	151.28	115	16.555	11.0706	48.29	1904	9.188	0.01	4	11.5	211.12
CO_2_	44.01	217.42	491.354	156.03	244	79.665	24.2846	54.72	See	below	2.99	1	0	214.61
N_2_O	44.013	215.9	489.65	155.94		66.9	22.255	59.9	See	Below	3.05		0	218.89

**Table entropy-21-00454-t002b:** (**b**)

Gas	MW	I_t_ × 10^−40^ (g·cm^2^)	@_t_/ħ	S_t_ (J·K^−1^)	I_rA_ × 10^−40^	I_rB_ (g·cm^2^)	I_rC_ × 10^40^	@_rA_/ħ	@_rB_/ħ	@_rC_/ħ	σ_r_	S_r_ (J·K^−1^)	Point Group
H_2_O	18.015	88.372	313.268	144.8	1.024	1.92	2.947	2.7533	3.7699	4.6709	2	43.74	C_2v_
H_2_S	34.08	167.18	430.867	152.75	2.667	3.076	5.845	4.4435	4.7721	6.5779	2	52.52	C_2v_
O_3_	47.998	235.45	511.336	157.02	7.877	62.865	70.9	7.6366	21.5796	22.9104	2	79.94	C_2v_
SO_2_	64.063	314.25	590.74	160.62	13.807	81.328	95.356	10.1103	24.5377	26.5697	2	84.58	C_2v_

Data obtained from Herzberg [12,13], Aylward and Findlay [15] or NIST website; values computed as described in Figure 2 using coding available at the Entropy site or directly from the corresponding author. Three moments of inertia are required for polyatomic molecules but only two equal moments for diatomics or linear molecules.

**Table entropy-21-00454-t002c:** (**c**)

H_2_O	Wave Number	x = hcν_i_/kT	S_vi_		CO_2_	Wave Number	x = hcν_i_/kT	S_vi_	Degen	∑S_vi_
A_1_	3652	17.6235	<0.0001		σ_g_^+^	1388	6.6981	0.079	1	0.079
A_1_	1595	7.697	0.0329		Π	667	3.2188	1.4547	2	2.9093
B_2_	3756	18.1254	<0.0001		σ_u_^+^	2349	11.3356	0.0012	1	0.0012
Total		Total	0.033						Total	2.9895
**H_2_S**		x = hcν_i_/kT	S_v_		**N_2_O**		x = hcν_i_/kT	S_v_		
A_1_	2615	12.6193	0.004		∑	2224	10.7324	0.0002		
A_1_	1183	5.7088	0.1856		∑	1285	6.2011	0.1216		
B_2_	2626	12.6723	<0.0001		Π	589	2.8423	1.4627		
		Total	0.186		Π	589	2.8423	1.4627		
							Total	3.0473		
**O_3_**		x = hcν_i_/kT			**SO_2_**		x = hcν_i_/kT			
A_1_	1110	5.3565	0.2504		A_1_	1151	5.5544	0.2117		
A_1_	705	3.4021	1.2561		A_1_	518	2.4997	2.5715		
B_2_	1042	5.0284	0.3301		B_2_	1352	6.5244	0.0919		
		Total	1.8367				Total	2.8751		

**Table entropy-21-00454-t003a:** (**a**)

Gas	MW	I_t_ × 10^40^ g·cm^2^	@_t_/ħ	S_t_ (J·K^−1^)	I_rA_ × 10^40^	I_rB_(g·cm^2^)	I_rC_ × 10^40^	@_rA_/ħ	@_rB_/ħ	@_rC_/ħ	σ_r_	S_r_(J·K^−1^)	Point Group
NH_3_	17.031	83.543	27.988	144.1	2.9638	2.9638	4.5176	4.6841	4.6841	5.783	3	48.36	*C_3v_*
	Vibrational	NH_3_	cm^−1^	Species	Wave number	x = hcν_i_/kT	S_v_						
				A_1_	3337	16.1034	0.0001						
				A_1_	950	4.5844	0.4785						
				E	3447	16.6343	0.0001						
				E	1627	7.8514	0.0287						
						Total	0.5074						

**Table entropy-21-00454-t003b:** (**b**)

Gas	MW Daltons	I_t_ × 10^40^ g·cm^2^	@_t_/ħ	S_t_(J·K^−1^)	I_rA_ × 10^40^	I_rB_(g·cm^2^)	I_rC_ × 10^40^	@_rA_/ħ	@_rB_/ħ	@_rC_/ħ	σ	S_r_(J·K^−1^)	Point Group
CH_4_	16.401	78.678	295.623	143.35	5.27	5.27	5.27	6.2461	6.2461	6.2461	12	42.263	*T_d_*
CFCl_3_	137.37	673.84	865.041	170.13	340.35	340.35	799.71	50.197	50.197	62.433	3	107.59	*C_3v_*
CF_2_Cl_2_	120.91	593.12	811.578	168.54	203.73	318	375.66	38.8364	48.5206	52.737	2	107.13	*C_2v_*
CF_3_Cl	104.46	512.41	754.336	166.71	146.32	251.58	251.58	32.9126	43.1566	43.156	3	99.745	*C∞v*
CH_4_	Wave number	x = hcν_i_/kT	S_v_	Degn.	S_v_		CFCl_3_	Wave Number	x = hcν_i_/kT	S_v_	Degn.	S_v_	
A	2914	14.063	0.0001	1	0.0001		A	1085	5.2359	0.2773	1	0.2733	
E	1526	7.364	0.0441	2	0.0882		A	535	2.5818	2.4105	1	2.4105	
T	3020	14.575	0.0001	3	0.0002		A	350	1.689	4.8792	1	4.8792	
T	1306	6.3034	0.1113	3	0.334		E	847	4.0874	0.7208	2	1.4416	
				Total	0.4225		E	394	1.9013	4.121	2	8.242	
							E	241	1.163	7.5121	2	15.024	
											Total	32.275	

**Table entropy-21-00454-t003c:** (**c**)

CF_2_Cl_2_ Band	cm^−1^	x = hcν_i_/kT	S_v_	CF_3_Cl	Band	cm^−1^	x = hcν_i_/kT	S_v_	Degeneracy	∑S_v_
A	1101	5.3131	0.2598		A	1105	5.3324	0.2556	1	0.2556
A	667	3.2188	1.4547		A	781	3.7689	0.9344	1	0.9344
A	458	2.2108	3.2297		A	476	2.297	3.0163	1	3.0163
A	262	1.2643	6.8968		E	1212	5.8488	0.1646	2	0.3293
A	322	1.5539	5.4387		E	563	2.7169	2.1667	2	4.3334
B	902	4.3528	0.5796		E	350	1.689	4.8792	2	9.7584
B	437	2.1088	3.4981						Total S	18.627
B	1159	5.593	0.2048							
B	446	2.1523	3.3804							
		Total S	24.945							

**Table 4 entropy-21-00454-t004:** Summary of total molar entropy terms.

Gas	St J·K^−1^	Sr J·K^−1^	Sv J·K^−1^	S J·K^−1^	Ref. [15]	IR Spectrum Vibration Wavelength Mm (Degeneracy Bracketed)
**H_2_O**	**144.8**		**0.033**	**188.6**		
CO_2_	155.94	54.72	2.99	213.6	214	4.257, 7.2046, 14.993
H_2_S	152.81	52.54	0.19	205.5	206	3.808, 3.824, 8.453
N_2_O	155.94	59.8	3.05	218.9	220	4.446, 7.782, 16.978
O_3_	157.02	79.94	1.84	238.8	239	9.009, 9.597, 14.184
SO_2_	160.62	84.58	2.875	248.1	248	7.396, 8.688, 19.305
NH_3_	144.1	48.36	0.507	193	192	2.901, 2.997, 6.146, 10.526
CH_4_	143.36	42.26	0.423	186	186	3.311(2), 3.432(3), 6.553, 7.657(3)
CFCl_3_	170.14	107.59	32.275	310	310	9.217, 11.806(2), 18.692, 25.381(2), 28.571, 41.494(2)
CF_2_Cl_2_	168.545	107.14	24.945	300.6	301	8.628, 9.083, 11.086, 14.999, 21.834, 22.421, 22.883, 31.056, 38.168
CF_3_Cl	166.721	99.75	18.627	285.1	286	8.251(2), 9.050, 12.804, 17.762(2), 21.008, 28.571(2)
O_2_	162.07	43.93	0.035	206	205	6.329
CO	150.31	47.19	0.0025	197.5	198	4.608

**Table 5 entropy-21-00454-t005:** Summary of total entropy, entropic energy in the real surface atmosphere at 298.15 K.

Gas	Pressure (Atm)	S_t_	S_r_	S_v_	S Total	∑S/Mole STP	J/Mole of Air/K	J Per m^3^
H_2_O	0.00775	185.29	43.74	0.033	229.1	188.6	529.37278	21,637.48
CO_2_	0.000397	215.51	54.72	2.99	273.2	213.6	32.337468	1,321.76
H_2_S	2 × 10^−10^	338.49	52.52	0.19	391.2	205.5	0.0000023	0.000003
N_2_O	0.000000325	280.24	59.9	3.05	342.2	218.9	0.0331588	1.355326
O_3_	2.66 × 10^−8^	302.13	79.94	1.84	383.9	238.8	0.0032865	0.134332
SO_2_	3 × 10^−10^	343.01	84.58	2.875	430.5	248.1	0.0000039	0.000159
NH_3_	5 × 10^−10^	322.23	48.36	0.507	371.1	193	0.0000553	0.00226
CH_4_	0.0000017	272.21	42.26	0.423	314.9	186	0.1596086	6.523810
CFCl_3_	2.6 × 10^−10^	350	107.59	32.28	489.9	310	0.000038	0.000006
CF_2_Cl_2_	5.5 × 10^−10^	345.9	107.14	24.945	478	300.6	0.0000784	0.000001
CF_3_Cl	1 × 10^−10^	353.08	99.75	18.627	471.5	285.1	0.0000141	0.000002
O_2_	0.2095	165.05	43.93	0.035	209	205.1	13,054.65	533,593.02
CO	0.00000015	279.58	47.19	0.0025	326.8	197.5	0.0146153	0.597382
NO	3 × 10^−10^	333.56	48.39	0.0087	382	199.6	0.0000034	0.000139
H_2_	0.0000005	238.11	12.7	-	250.8	130.2	0.037388	1.52819
N_2_	0.78084	152.45	41.27	-	193.7	191.7	45,094.80	1,843,195.93
Ar	0.00934	193.7	-	-	193.7	154.8	539.400466	0.0442339
							**Total**	**2,399,758.37**

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
