# Peer review of "A Simple Method to Estimate Entropy and Free Energy of Atmospheric Gases from Their Action"

_entropy, 2019, doi:10.3390/e21050454_

Round 1
Reviewer 1 Report
All suggestions have been marked within the manuscript pdf file.

Author Response
Thank your very much for your suggestions. Our responses to your comments is attached here.

Reviewer 2 Report
This paper considers a theoretical model, based on the statistical mechanics, for calculating the total entropy of atmospheric gases. Thermodynamic properties are estimated using algorithms based on molecular and physical properties simplifying the estimation process as it doesn’t need to use tables of standard values and the links among heat, workmodelingand other properties. According to the authors, this approach is more realistic and can provide a better understanding of some properties showed by natural or complex systems. Applications can be done for modelling atmosphere and authors recommend the widespread of this easy methodology.
The perspective shown in this paper is interesting and it’s based on easy algorithms and offers a new approach for calculating atmospheric gases’ entropy. Explanations are well formulated and easy to understand.
It is well written, even though sometimes the language used is not as scientific as it should be, Entropy is a scientific journal, not a textbook. Besides, the format is not fully coherent: some equations are numbered and other aren’t, references are not always written following the same rules (then, without following Journal recommendations -at least in some cases).
Equation shown in line 81 should be numbered, as well as equations in lines 85, 87, 89, 155, 203, 298, 314, 327, 361, 362, 363, 364, 390, 665-667, 672, 683.
Regarding the Sackur-Tetrode equation, there is a lack of explanation about how the authors move from a theory based on monoatomic gases to polyatomic gases.
All the equations mentioned between lines 129 to 153 should be written as an equation and not as text explained in the body of the paper. It’s really confusing: 1/2mv2=3/2kT (line 136, as an example) or Itwt2=3kT (line 138) and so on. Something similar happens in other paragraphs of the paper, lines 491 to 496 or lines 497 to 502, lines 506 to 510, line 608, lines 625 to 632, lines 687 to 695.
Tables, all around the paper, have a too short title, with no explanation about the meaning of the different columns/rows.
Besides, each table is different (Table 2 shows a “Microsoft Word” look, but table 3a is a list of numbers).
In some tables (see table 3c, table 4a, and table 4b):
· there are empty files and rows,
· the columns are different at some point of the table,
· some column titles are in bold and others are not,
· units are not written between brackets,
· references are mentioned in different ways (line 257: page used should be indicated in the bibliography, not in the table),
· parameters are not explained in the text.
In table 5 (line 434) the meaning of the numbers between brackets is disappeared (CH4 3.311 (2)?).
Line 472 to 474: which is the meaning of “a 5-6 very low proportion […]? This sentence has no sense.
Line 475, shown in table 4 but in which one, table 4a or table 4b?
Line 482, it is not written which formula (equation?) is used nor from which column it comes.
Tables 3a-3b-3c and tables 4a-4b sometimes are table 3 or table 4 and others aren’t. Please be consistent and, considering there are several tables, it should be indicated which of them is the mentioned (line 475, 506).
Line 530: explain the sentence and why it is written ppmv between brackets. From my knowledge, ppmv is a concentration unit (part per million in volume), and I cannot see any column referred to concentrations in Table 5.
Line 715: it looks like a lost sentence.
The way to mention the references varies along the text, it should be always the same: [number].
It could be interesting to extend the conclusions, with more explanations. The paper offers a link between theory and practice that can be applied easily, hence this point should be remarked.

Author Response
Thank your very much for your suggestions. Our responses to your comments is as following:
Authors’ responses:
This paper considers a theoretical model, based on the statistical mechanics, for calculating the total entropy of atmospheric gases. Thermodynamic properties are estimated using algorithms based on molecular and physical properties simplifying the estimation process as it doesn’t need to use tables of standard values and the links among heat, work and other properties. According to authors, this approach is more realistic and can provide a better understanding of some properties showed by natural or complex systems. Applications can be done for modelling atmosphere and authors recommend the widespread of this easy methodology.
The perspective shown in this paper is interesting and it’s based on easy algorithms and offers a new approach for calculating atmospheric gases’ entropy. Explanations are well formulated and easy to understand.
It is well written, even though sometimes the language used is not as scientific as it should be, considered Entropy is a scientific journal and it is not a textbook. Besides, the format is not fully coherent: some equations are numbered and other aren’t, references are not always written following the same rules (then, without following Journal recommendations -at least in some cases).
Responses: Given the novelty of the approach, there are few recent references in the paper. Well-known text books are referred to instead.
Equation shown in line 81 should be numbered, as well as equations in lines 85, 87, 89, 155, 203, 298, 314, 327, 361, 362, 363, 364, 390, 665-667, 672, 683.
Responses: All equations are now numbered as recommended
Regarding the Sackur-Tetrode equation there is a lack of explanation about how the authors move from a theory based on monoatomic gases to polyatomic gases.
Responses: Two sentences in explanation are inserted at line 196
All the equations mentioned between lines 129 to 153 should be written as equation and not as text explained in the body of the paper. It’s really confusing: 1/2mv2=3/2kT (line 136, as an example) or Itwt2=3kT (line 138) and so on. Something similar happens in other paragraphs of the paper, lines 491 to 496 or lines 497 to 502, lines 506 to 510, line 608, lines 625 to 632, lines 687 to 695.
Responses: These items have been extracted as equations
Tables, all around the paper, have a too short title, with no explanation about the meaning of the different columns/rows.
Besides, each table is different (table 2 shows a “Microsoft Word” look, but table 3a is a list of numbers).
Responses: All tables have been revised and are now prepared in Microsoft mode. The criticisms below have been accepted and suitable corrections made in all cases. Units are now in brackets.
In some tables (see table 3c, table 4a and table 4b):
· there are empty files and rows,
· the columns are different at some point of the table,
· some column titles are in bold and others are not,
· units are not written between brackets,
· references are mentioned in different ways (line 257: page used should be indicated in the bibliography, not in the table),
· parameters are not explained in the text.
In table 5 (line 434) the meaning of the numbers between brackets is disappeared (CH4 3.311 (2)?).
Responses: The bracket is explained in the title.
Line 472 to 474: which is the meaning of “a 5-6 very low proportion […]? This sentence has no sense.
Line 475, shown in table 4 but in which one, table 4a or table 4b?
Line 482, it is not written which formula (equation?) is used nor from which column it comes.
Tables 3a-3b-3c and tables 4a-4b sometimes are table 3 or table 4 and others aren’t. Please be consistent and, considering there are several tables, it should be indicated which of them is the mentioned (line 475, 506).
Responses: Necessary corrections have been made.
Line 530: explain the sentence and why it is written ppmv between brackets. From my knowledge, ppmv is a concentration unit (part per million in volume), and I cannot see any column referred to concentrations in Table 5.
Responses: This pressure is now given in atm.
Line 715: it looks like a lost sentence.
The way to mention the references varies along the text, it should be always the same: [number].
Responses: Corrected.
It could be interesting to extend the conclusions, with more explanations. The paper offers a link between theory and practice that can be applied easily, hence this point should be remarked.
Responses: Two extra sentences have been added to the conclusions.
